# Positive Orientation and Strategies for Coping with Stress as Predictors of Professional Burnout among Polish Nurses

**DOI:** 10.3390/ijerph16214264

**Published:** 2019-11-02

**Authors:** Ewa Kupcewicz, Marcin Jóźwik

**Affiliations:** 1Department of Nursing, Faculty of Health Sciences, Collegium Medicum, University of Warmia and Mazury in Olsztyn, 14 C Zolnierska Street, 10-719 Olsztyn, Poland; 2Department of Gynecology and Obstetrics, Faculty of Medicine, Collegium Medicum, University of Warmia and Mazury in Olsztyn, 44 Niepodleglosci Street, 10-045 Olsztyn, Poland; marcin.jozwik@uwm.edu.pl

**Keywords:** positive orientation, stress, burnout, nurses

## Abstract

*Background:* A psychological resource, positive orientation, may moderate stress and protect nurses from burnout. The purpose of this study was to determine the role of positive orientation and stress-coping strategies in predicting professional burnout among Polish nurses. *Methods:* A total of 1806 nurses employed in 23 hospitals in northeastern Poland participated in the study. The study used a standardized Positive Orientation Scale, Mini-COPE, and the Copenhagen Burnout Inventory. *Results:* On the stenographic scale, 21.8% of nurses had high levels of positive orientation, 46.8% were average, and 31.9% had low positive orientation. A total of 21.1% of respondents reported personal burnout. Most nurses reported work-related burnout (27%) and burnout in contacts with patients (28.4%) With increasing levels of positive orientation, nurses more often used adaptive strategies that focus on a problem and emotions. Personal burnout accounted for 16% of the variance of the dependent variable (β = −0.32; *R*^2^ = 0.16), which was slightly lower than work-related burnout (10% (β = −0.23; *R*^2^ = 0.10)), and burnout in contacts with patients (9% (β = −0.22; *R*^2^ = 0.09)). *Conclusion:* The reduced level of positive orientation proved to be the main determinant of the professional burnout of Polish nurses. Burnout-prophylactic programs should be geared towards developing individual psychological resources, including positive orientation and the acquisition of effective stress-coping skills.

## 1. Introduction

In the course of performing work-related tasks, nurses are exposed to many harmful, cumbersome, and dangerous factors, including emotional and interpersonal stressors [1]. As early as the 1950s, M.S. Schwartz and G.T. Will described the case of a nurse working in a psychiatric ward who began to lose energy and motivation to work. The nurse felt mental and physical fatigue, exhaustion, helplessness, skepticism, and lack of joy in performing nursing tasks as a result of intense tension at work caused by various stressors [2]. In subsequent studies, it has been shown that the observed symptoms were evidence of burnout [3,4,5].

In Europe, Danish researchers from Copenhagen lead by T.S. Kristensen identified three components of burnout: personal burnout, work-related burnout, and burnout in contacts with patients [6,7]. Personal burnout was defined as “physical and psychological fatigue and exhaustion experienced by a given person”, work-related burnout by the researchers referred to “physical and mental fatigue or exhaustion experienced by a person while performing work”, whereas burnout in contacts with patients was defined as “a physical and mental fatigue or exhaustion experienced by a person when dealing with patients” [6,7].

An overview of the available literature clearly highlights the variety of factors associated with burnout syndrome [1]. Most often, they can be classified into three groups: organizational factors, socio-demographic and lifestyle factors, and personality variables and coping strategies [1,3,8,9,10]. Worth noting is the study by Manzano-García and Ayala with a group of 40 experts from 6 countries. The study identified potentially important factors in explaining burnout in nursing that have been insufficiently studied or ignored. Among the many analyzed factors, the authors emphasized the importance of personal factors and personal resources in the process of professional burnout [1]. Other studies show that personal (psychological) resources, often considered to be stress moderators, such as optimism, high self-esteem, effectiveness, and the ability to build relationships with other people, can facilitate an individual to cope better with stressors. This effect takes place even if the encountered events are evaluated in the categories of stress, but an individual does not experience the negative effects or experiences less of the negative effects in the process [11,12,13]. The authors of this paper argue that there is a need to continue research into the theory of positive orientation, as positive orientation refers to the components of mental well-being in everyday life and may play a significant role in the professional burnout of nurses.

Gian Vittorio Caprara, one of the most eminent Italian psychologists, formulated the concept of positive orientation, which fits into the positive psychology of the last quarter century [14]. Positive orientation is defined by Caprara as “a fundamental tendency to notice and to attach importance to the positive aspects of life, experience and self” [15,16,17,18]. The results of the Italian and Canadian studies confirmed that positive orientation is a latent (hidden) higher order variable that combines three components: self-esteem, optimism, and satisfaction from life [16,17,18]. In the literature, it is emphasized that positive orientation is largely responsible for adaptive functioning and is motivational in the actions of individuals [17,18,19,20]. It signifies “natural inclinations towards positive self-esteem, high satisfaction in life, and a high evaluation of the chances of achieving goals, which translates into commitment to life’s aspirations and a high quality of life” [17,18,19,20]. Alessandri et al. investigated the relationship between positive orientation and behavior in organizations. As it turned out, positive orientation was significantly associated with selected organizational behaviors that brought immediate benefits to the organization as a whole, and altruistic behaviors aimed at benefiting individuals [21]. The results of Polish research have shown that positive orientation and self-efficacy are significantly connected but distinct variables [22]. In studies conducted by Miciuk et al., 200 students showed moderate correlations between positive orientation and a generalized sense of effectiveness, positive affect, sense of meaning, and purpose in life. In other studies, there was a positive relationship between gratitude, self-esteem, and positive orientation [23]. The above empirical data draw a picture of potential relationships between positive orientation and functioning [24].

Over the last few years, increasing research has been undertaken on human activities to deal with stressful events, referred to as the concept of coping with stress [11]. Most of the research refers to the transactional approach of stress, formulated by R. Lazarus and S. Folkman, that stress is a distortion of the relationship between the individual and the environment [11,25]. Researchers have identified an instrumental coping function in dealing with problems (focused and regulatory coping strategies) linked to coping strategies focused on emotion [25]. Subsequent studies have allowed us to distinguish between disposable and situational coping [26]. In the case of disposable coping, there is a style of cognition understood as a relatively constant, individual-specific tendency that determines the course of coping with stress [27]. Style has the status of a personality variable and is an entity-specific repertoire of strategies for coping with stressful situations [26]. Referring to the transactional concept of stress, N. Endler and J. Parker identified three types of coping, two focused on task and emotion, and the other focused on avoidance [27]. The attempt to combine coping, understood as style and strategy, was presented by C.S. Carver et al. [28]. Referring to the Lazarus theory and the behavioral self-regulation model, they have proposed several coping strategies that can reflect both a fixed tendency to cope in a specific way and those used in a particular stress situation, referred to as situational coping [11,28].

According to the theoretical assumptions of the analyzed variables, for the authors of this study, it was interesting to what extent those components will affect the process of professional burnout in nurses.

By scientific inquiry, the following research problems have been formulated:
Do selected sociodemographic variables, that is, age, work experience, and financial situation significantly differentiate a positive orientation among nurses?What is the relationship between the level of positive orientation of nurses and their strategies of coping with stress and personal burnout, work-related burnout, and burnout in contact with patients?What is the role of positive orientation, coping strategies, and selected sociodemographic variables of nurses in the prediction of personal burnout, work-related burnout, and burnout in contact with patients?

It can be expected that both positive orientation and the strategies employed by nurses to deal with stress will be the moderators of increased burnout, in line with the model presented in Figure 1.

## 2. Materials and Methods

### 2.1. Settings and Design

Empirical material was collected from June 2013 to January 2015 in 23 hospitals in northeastern Poland: (7) Olsztyn, (2) Elbląg, (2) Ełk, (1) Działdowo, (1) Nowe Miasto Lubawskie, (1) Kętrzyn, (1) Szczytno, (1) Biskupiec, (1) Iława, (1) America, (1) Ostróda, (1) Pisz, (1) Mrągowo, (1) Nidzica, and (1) Giżycko. The study was held at the workplace of the nurses, with the consent of the directors of the hospitals. The main criteria for inclusion in the sample were at least one year of seniority in the nursing profession and an informed consent to participate in the study. The prepared sets of questionnaires were delivered to the hospitals participating in the research project by one of the researchers. The study was anonymous. Respondents were informed about the purpose of the survey, and had the opportunity to ask questions and get clarification. They also had the right to withdraw from the study at any time with no need to give reasons. Upon granting consent for participation in the study, respondents received a set of questionnaires to complete, which were returned within five days from all the hospitals. The study period was about 30 min. In total, 2885 questionnaire forms were distributed among medical personnel (nurses/midwives). Upon collection of data and elimination of faulty questionnaires filled in by nurses, 1806 (62.6%) questionnaires were qualified for further analyses. The return percentage varied depending on the healthcare facility and ranged from 25.3% to 87.5%.

The study was conducted in line with the principles stated in the Declaration of Helsinki. The Senate Research Ethics Committee of Olsztyn University College J. Rusiecki, Poland, issued a positive opinion (no. 11/2016) on the ethical aspects of the research project. The study meets the criteria of a cross-sectional study [29]. The empirical data used in the paper constitutes a part of a larger research project conducted in Polish hospitals.

### 2.2. Participants

The mean age of participants was 44.69 (± 7.96). The most numerous group were nurses aged 41–50 years (*n* = 862; 47.7%). The vast majority lived in the city (79.3%; *n* = 1432). The average work experience in nursing was 22.5 years (± 9.5). Three-quarters of the respondents (75.7%) worked in a two-shift system, including night-time shifts. A third of the respondents (36.8%; *n* = 665) had secondary education (graduated medical high school), 20.5% (*n* = 371) had post-secondary education, 25.4% (*n* = 458) had first degree higher education (bachelor’s degree), and 11.8% (*n* = 214) had second degree higher education (master’s degree). Most of the respondents were married or had a partner (78.3%; *n* = 1414), only 15.1% (*n* = 272) did not have children. In the study group, 104 people (5.8%) had a disability rating. The financial situation of nurses was varied. A total of 42.9% (*n* = 774) claimed that their financial situation was at a sufficient level, 30.4% (*n* = 567) claimed a good financial situation, and 18.6% (*n* = 336) indicated that it was poor. The largest group was represented by nurses originating from wards/clinics offering conservative treatment (*n* = 804; 44.5%); a slightly smaller group contained nurses from treatment units (*n* = 574; 31.8%); a small group consisted of nurses from intensive care, anaesthesiology, operating rooms (*n* = 245; 13.6%); and the remainder consisted of other units, such as counselling units and diagnostic offices (*n* = 183; 10.1%).

### 2.3. Research Instruments

A diagnostic survey was used as a research method, and an own-design questionnaire and three standardized research tools were used for data collection: A Positive Orientation Scale by G.V. Caprara et al., in Polish adaptation by M. Łaguna, P. Oles, D. Filipiuk [19]; Mini-COPE questionnaire by C.S. Carver, in the Polish adaptation by Z. Juczyński and N. Ogińska-Bulik [12]; and the Copenhagen Burnout Inventory (CBI) by T.S. Kristensen et al. [6] were used. Sociodemographic variables, such as gender, age, place of residence, marital status, education, financial situation, and length of service, were identified using an own design questionnaire.

#### 2.3.1. The Polish Adaptation of the Positive Orientation Scale

The Polish version of the Positive Orientation Scale (P-Scale) is made of eight statements, all of which are diagnostic. A respondent refers to each statement, indicating to what extent they agree with each statement. Answers are given on a five-point scale from 1—“I definitely disagree”, to 5—“I strongly agree”. The result is the sum of the points, which is an indicator of the general level of positive orientation. The range of achievable results ranges from 8 to 40 points. The higher the score, the higher the level of positive orientation. The scale has good psychometric properties—Cronbach α coefficient values were in the range of 0.77 to 0.84 and indicate a satisfactory consistency of the method. Standardized tests have stenographic standards developed. The overall index of the level of positive orientation is transformed into standardized units, which are interpreted according to the characteristics of the stenographic scale. It contains 10 units, and the scale jump equals 1 sten. Results in the range of 1 to 4 stens are considered low results, those ranging from 5 to 6 stens are average, and between 7 and 10 stens is considered high [18].

#### 2.3.2. The Polish Adaptation of the Mini-COPE

The Polish version of the Mini-COPE inventory was used to measure coping strategies. It is a shortened version of the Multimodal Inventory for Measurement of Coping with Stress-COPE (by C.S. Carver, M.F. Scheier, and J.K. Weintraub, Polish adaptation by Z. Juczyński and N. Ogińska-Bulik) and measures coping in terms of disposition. It consists of 28 statements that are part of 14 strategies for coping with stress, including active coping, planning, positive revalidation, acceptance, sense of humor, turn to religion, seeking emotional support, seeking instrumental support, taking care of something else, denial, discharge, use of psychoactive substances, cessation of activities, and self-blaming. There are two theorems for each strategy. The tested respondent refers to each statement by marking one possible answer on a four-point scale from 0 (“I almost never do so”) to 3 (“I almost always do so”). The obtained psychometric properties are satisfactory, the half reliability for the 14 strategies is 0.86 [11].

#### 2.3.3. Copenhagen Burnout Inventory (CBI)

The Copenhagen Burnout Inventory contains 19 questions that relate to the attitude to work and the feelings involved. The inventory measures three dimensions of professional burnout: personal burnout (the dimension of a person’s physical and mental fatigue and exhaustion assessment, six questions); work-related burnout (physical and mental fatigue, seven questions); and burnout in contacts with patients (a dimension describing the physical and mental fatigue or exhaustion experienced by a person in contact with patients, six questions). Respondents can respond on a five-point scale by selecting one of the following possible answers: always or very high—100 points, often or to a large extent—75 points, occasionally or somewhat—50 points, rarely or to a small extent—25 point, never/almost never or to a very small degree—0 points. In the case of one work-related burnout question, according to the authors’ recommendations, the inverse score should always be used: always—0 points, often—25 points, occasionally—50 points, rarely—75 points, never/almost never—100 points. The total score for each of the three burnout subscales is the mean value obtained from the individual parts. The psychometric properties of the CBI questionnaire are satisfactory; the α-Cronbach coefficient for evaluating individual burnout dimensions is very high and ranges from 0.85 to 0.87 [6].

### 2.4. Statistical Analysis

The data generated during an a posteriori study were subjected to statistical analysis using the Polish version of STATISTICA 13 (TIBCO, Palo Alto, CA, USA). The following statistical methods were used: positional measures and measures of variability to describe the structure of the population; the stenographic scale; the chi-squared independence test (χ^2^) to evaluate the dependence of non-measurable (*x*, *y*) qualities; the Kruskal–Wallis (H) test, to compare the distribution of the variables for the number of trials >2; the post-hoc test for multiple mean ranks for all trials; the Spearman’s (R) rank correlation coefficient, which allowed a determination of the relationships between variables with a value in the closed range of [−1, 1]; and a multiple regression analysis to construct a random variable estimation model from explanatory variables [30]. Statistical tests were performed at a significance level of *p* < 0.05.

## 3. Results

Statistical analysis showed that the overall mean positive orientation index determined with the Positive Orientation Scale (P-Scale) in the group of nurses was 28.90 (±4.15) points. After transforming into standardized units, according to stenographic features, 21.3% (*n* = 384) of nurses had a high level of positive orientation, whereas 46.8% (*n* = 846) had average, and 31.9% (*n* = 576) had a low level of positive orientation. The mean positive orientation result obtained in the stenographic scale was 5 stens and reflected the average level of positive orientation in the study group. In the course of the analysis using the chi-square independence test (χ^2^), there was a statistically significant relationship between positive orientation and the financial situation of nurses (χ^2^ = 155.23; *p* < 0.001). The differences between the independent groups due to the values of the analyzed variables were confirmed by a Kruskal–Wallis non-parametric test (H = 152.83; *p* < 0.001; Table 1). In subsequent analysis using multiple post-hoc comparisons of mean ranks for all trials, statistically significant (*p* < 0.001) intergroup differences were found. Nurses who thought they had a very good financial situation were characterized as having a higher level of positive orientation (31.99 ± 4.52) than those who indicated that their financial situation was at a sufficient level (28.58 ± 3.92) or at a poor level (27.18 ± 4.05).

Further analysis confirmed that a positive orientation correlated positively with the financial situation of nurses. Interpretation of the relationship between variables was based on Guilford’s classification [30]. It was considered to be a statistically significant weak correlation (R = 0.28; *p* < 0.001), suggesting that as the level of positive orientation increased, the financial situation of nurses increased (Table 1). In the collected data, there was no significant relationship between positive orientation and the age and work experience of the respondents. This study sought to investigate the relationship between positive orientation and stress coping strategies used by nurses using Spearman’s rank correlation coefficient (R).

The analysis of the data presented in Table 2 shows that the positive orientation was weak in relation to strategies for coping with stress among nurses. The results showed statistically significant positive correlations between positive orientation and coping strategies, such as planning (R = 0.27; *p* < 0.001), seeking emotional support (R = 0.27; *p* < 0.001), positive revalidation (R = 0.22; *p* < 0.001), acceptance (R = 0.19; *p* < 0.001), and seeking instrumental support (R = 0.19; *p* < 0.001). In addition, positive orientation had a negative relationship with helpless coping strategies, such as self-blaming (R = −0.26; *p* < 0.001) and cessation of actions (R = −0.21; *p* < 0.001). It may be concluded that with the increase in the level of positive orientation, nurses more often use adaptive strategies, which are focused on the problem and emotions. On the other hand, a low level of positive orientation contributes to the more frequent use of non-adaptive coping strategies. Other strategies of coping with stress (with the exception of the strategy referred to as turn to religion) have also been observed, but their correlation was weak, albeit statistically significant (all at *p* < 0.001; Table 2). In the course of further work, levels of burnout among nurses were analyzed. After calculating the average scores for individual burnout components, the criteria for no burnout (mean ≤45 points), burnout hazard (average 45 < points ≤ 55), and presence of burnout (mean >55 points) were established [31]. Half of the respondents (50.7%; *n* = 915) reported the absence of personal burnout, slightly less (46%; *n* = 832) reported no work-related burnout or burnout in contact with patients (44.6%; *n* = 807). More than a quarter of respondents claimed that they were at risk of burnout in all components when considering the following values: personal burnout 28.2% (*n* = 510), work-related burnout 27% (*n* = 487), and burnout in contacts with patients 27% (*n* = 487). There were 21.1% (*n* = 381) of subjects reporting personal burnout, whereas significantly more nurses reported work-related burnout (27%, *n* = 487) and burnout in contact with patients (28.4%; *n* = 512). The next step in the analysis of the research findings was to determine the relationship between positive orientation, stress coping strategies, age, work experience, financial situation, and three dimensions of professional burnout—personal burnout, work-related burnout, and burnout in contacts with patients. The data presented in Table 3 show the average negative correlation between positive orientation and personal burnout (R = −0.39; *p* < 0.001), work-related burnout (R = −0.31; *p* < 0.001), and burnout in contacts with patients (R = −0.30; *p* < 0.001). In interpreting this data, it can be concluded that the level of professional burnout of nurses in all three components decreased as the level of positive orientation increased. This may mean better social functioning of nurses with a higher level of positive orientation. In this way our studies show that positive orientation plays a protective role against burnout. According to the model presented in Figure 1, it was confirmed that there was a link between professional burnout and the preferred stress coping strategies. The results showed a positive correlation between personal burnout and non-adaptive coping strategies. They indicated that interdependence of coping strategies, such as denial (R = 0.16; *p* < 0.001), discharge (R = 0.16; *p* < 0.001), use of psychoactive substances (R = 0.13; *p* < 0.001), cessation of actions (R = 0.20; *p* < 0.001), self-blaming (R = 0.19; *p* < 0.001), and the presence of symptoms of burnout in nurses was weak. There was also a statistically significant weak correlation between personal burnout and three adaptive strategies for coping with stress. With the increased use of adaptive coping strategies, such as active coping (R = −0.17; *p* < 0.001), planning (R = −0.13; *p* < 0.001), and seeking emotional support (R = −0.15; *p* < 0.001), there was a decrease in the symptoms of personal burnout. Two strategies, such as positive revalidation (R = 0.08; *p* < 0.001) and acceptance (R = −0.09; *p* < 0.001) showed a weak correlation with personal burnout. These studies show that nurses use a variety of strategies in stressful work situations, which may indicate some flexibility in coping. In addition, the level of burnout at work was weak and significantly negatively associated with active strategies such as planning (R = −0.13; *p* < 0.001), active coping (R = −0.18; *p* < 0.001), seeking emotional support (R = −0.13; *p* < 0.001), and acceptance (R = −0.10; *p* < 0.001). The established correlation between variables showed that by increasing the frequency of adaptation strategies, there was a decrease in symptoms associated with work-related burnout. There was also a significant negative correlation with the strategy described as positive revalidation (R = −0.06; *p* < 0.001). The obtained correlation coefficients showed a significant weak positive relationship between work-related burnout and ineffective strategies, and non-adaptive strategies, cessation of actions (R = 0.18; *p* < 0.001), denial (R = 0.14; *p* < 0.001), discharge (R = 0.11; *p* < 0.001), use of psychoactive substances (R = 0.11; *p* < 0.001), and self-blaming (R = 0.11; *p* < 0.001). In this case, it can be concluded that the increase in burnout symptoms associated with work is related to the more frequent use of ineffective coping strategies. The analysis of data presented in Table 3 shows that nurses use a variety of coping strategies to deal with the difficulties and obstacles that arise in contacts with patients. There was a weak correlation between burnout in contacts with patient symptoms and the use of adaptive coping with stress strategies. The detected relationship was negative. It can be concluded that with an increased frequency of adaptive coping strategies such as active coping (R = −0.15; *p* < 0.001), planning (R = −0.13; *p* < 0.001), acceptance (R = −0.10; *p* < 0.001), and seeking emotional support (R = −0.14; *p* < 0.001), the symptoms of burnout experienced in contacts with patients decreased. This is a very important finding, as due to communication fulfilling the primary clinical function in working with patients, it thus has great therapeutic potential and forms the basis of the medical care model. Positive correlations were also found between age, work experience, and the severity of burnout in all three components. The strength of the relationship between the variables was weak. There was a negative correlation between the financial situation of nurses and personal burnout (R = −0.23; *p* < 0.001), work-related burnout (R = −0.22; *p* < 0.001), and burnout in contacts with patients (R = −0.21; *p* < 0.001). By interpreting this data, it can be concluded that the level of symptoms of burnout in nurses decreases with the change (increase) of the financial situation in favor of nurses.

In the course of further work, a multiple regression analysis was performed to identify the predictors having a real impact on the level of professional burnout among nurses. Positive orientation and stress coping strategies were adopted for the explanatory variables, and the age, work experience, and financial status of the respondents were taken into consideration. It was found that age and part of the coping strategies did not significantly affect the regression model.

As shown in Table 4, the predictors of personal burnout included seven variables, explaining a total of 21% variability. The largest share (16%) in the personal burnout prediction was positive orientation, which had a negative value (β = −0.32; *R*^2^ = 0.16). This means that with a lower level of positive orientation, a nurse feels more physically and mentally fatigued and exhausted. The other variables explained only 5% of the variance of the dependent variable, so they did not play a significant role in the burnout predictions. The data in Table 5 indicate that in predicting the intensity of work-related burnout, six variables were involved, explaining a total of 16% of the variability of results. The largest share was positive orientation, which explained 10% of the variance of the dependent variable (β = −0.23; *R*^2^ = 0.10). The negative value of the regression coefficient describes a negative correlation. It can be concluded that the more the level of positive orientation decreased, the greater the level of physical and mental fatigue or exhaustion experienced by nurses while performing work. Financial situation and work experience played a minor role in the prediction of work-related burnout and their share did not exceed 4%. The three other coping strategies, such as cessation of actions, active coping, and discharge, did not play a significant role in the prediction of work-related burnout.

Seven variables turned out to be predictors of nurse burnout in contacts with patients, and together accounted for 14% of the variability (Table 6). The largest share predicting burnout in contacts with patients was associated with positive orientation (9%), with a negative correlation (β = −0.22; *R*^2^ = 0.09). As in the two previous models, the lower the level of positive orientation, the more symptoms of professional burnout were present in contacts with patients. Financial situation was also proven to be a predictor of burnout in contacts with patients, but its contribution to the predictor of this variable was not significant (2%). Other variables, such as the strategy of cessation, work experience, active coping, discharge, and acceptance, did not play a significant role in the prediction of this dimension of professional burnout.

## 4. Discussion

In our own research, empirical data outlined a picture of potential interdependencies between positive orientation, stress coping strategies, and burnout among Polish nurses. One of the characteristics of this study was the inclusion of positive orientation as a variable that integrates three components: self-esteem, optimism, and satisfaction from life [16,17,18]. To date, the role of positive orientation in the prophylaxis of occupational burnout in nurses has not been taken into account in Polish studies. The overall average positive orientation indicator score of 28.90 (± 4.15) obtained by the surveyed nurses was close to the average results of the Polish standardization tests in adults (30.21 ± 4.73) and students (29.19 ± 4.55) and in other studies of women (29.74 ± 4.84) [18,23]. This indicates that, despite experiencing many symptoms of burnout, nurses do not differ in the level of positive orientation from the other study groups. No correlation was found between positive orientation and the age and work experience of the respondents. Also, in studies conducted by Oles et al. with a group of 672 people, the age of the subjects did not influence the relationship between positive orientation and self-efficacy [22]. Alessandri et al. published the results of a four-year longitudinal study to identify developmental changes in positive orientation and to analyze the links between positive orientation and personal and social adaptation in early adulthood [32]. The researchers showed that the level of positive orientation did not change significantly in the study population during the four-year period. The higher the level of positive orientation, the more often they experienced positive emotions and indicated a better quality of perceived interpersonal relationships [18,19]. As the results of the study show, positive orientation correlated positively with the financial situation of nurses. Nurses with a higher level of positive orientation are better at dealing with material and financial needs. In the study group, 68.1% of nurses had average and high levels of positive orientation, whereas 31.9% of them rated the level of positive orientation as low. On the other hand, about one in four respondents said that they experienced symptoms of burnout in the three dimensions tested by the Copenhagen Burnout Inventory (mostly in contacts with patients). In the light of our own research, positive orientation was a variable that negatively correlated with the three components of burnout, which may prove to be an important personal resource. From the review of previous research, it is clear that positive orientation as a personal resource is adaptive. It allows for better coping with obstacles and difficulties, as well as being a motivation, especially when the individual is planning successive projects and designing their role in those projects [17,18,20].

To better understand the context of factors protecting the nurses against professional burnout, the authors of this study also referred to research on stress coping strategies in the workplace. Many researchers have clearly indicated in their research that professional burnout is associated with high levels of stress, and coping strategies can act as a factor in preventing professional burnout [5,33,34].

Garros et al. observed that active coping and social support were relevant in predicting the burnout dimensions. Lack of personal accomplishments and depersonalization were affected by active coping in an inverse temporal way [10]. Hamaideh et al., in a study conducted in a group of 464 nurses in 13 Jordanian hospitals, demonstrated that the most prevalent stressors among nurses from Jordan were heavy workload and having to deal with the issue of death [35]. In other studies, six main types of strategies for coping with occupational stress were found during data analysis: seeking help, self-controlling, situational control of conditions, preventive monitoring of the situation, spiritual coping, avoidance, and escape [36]. McTiernan and McDonald found that workload, organizational structures/processes, and lack of resources were identified as the main stressors [37]. Wilkinson, on the basis of literature review, found “the main causes of stress in the workplace for emergency and trauma nurses. These stressors include work demands and lack of time, lack of managerial support, patient aggression and violence, and staff exposure to traumatic events. Their effects on nurses include burnout, compassion fatigue, somatic complaints, mental health problems and difficulties in life outside work” [38]. In a study conducted with a group of 314 nurses in Taiwan, it was found that higher levels of proactive coping behaviors and optimism were associated with lower levels of burnout. Optimism was found to have the strongest correlation with decreased personal accomplishment and burnout [39]. Geuens et al. found that nurses with a D-type personality were five times more susceptible to stress than those with other personality types, regardless of organizational and work-related factors [40].

The obtained results of the authors’ own research indicate an important role of positive orientation in the selection of coping strategies in stressful situations. The interviewed nurses with increased positive orientation were more likely to use both problem-focused strategies (such as planning and active coping) and strategies that reduce stress and negative emotions (such as positive revalidation, seeking emotional support, and acceptance). These strategies, by stimulating positive emotions, play an adaptive role. The use of non-adaptive strategies that demonstrate helplessness, such as blaming and cessation of actions by nurses, among others, results from a low level of positive orientation. As shown by the results of this study, nurses use both active stress management strategies in their work, as well as less effective strategies, which may also prove to provide a degree of flexibility in coping. These results are in line with the previously cited sources on stress and coping strategies [33,34,35,36,37,38].

In the study group, the relationship between positive orientation as a personal resource and symptoms of burnout can be both direct and indirect, as illustrated in Figure 1. This indicates that a low level of positive orientation directly amplifies burnout symptoms, and indirectly increases the likelihood of nurses undertaking non-adaptive coping strategies, which then contribute to an increase in symptoms of burnout. In the search for the predictors of professional burnout, a low positive orientation showed the highest predictive power. Referring to Caprara’s positive orientation definition and data analysis, nurses with a low level of positive orientation who are experiencing burnout may find it difficult to perceive the positive aspects of life and are probably not drawing satisfaction from life [15].

Emotional exhaustion results in nurses feeling overwhelmed with work to such extent that they are unable to meet work requirements and cannot cooperate with others [41]. The authors of the paper focused on the consequences of professional burnout, which include, as confirmed by other researchers, increased nurse turnover indices, low work efficiency, and low involvement in patients’ safety [42]. As proven by many studies, the undertaking of multi-dimensional and interdisciplinary prophylactic activities in professional burnout among nurses should become an important element in human resource management in the healthcare system [42,43,44,45].

The presented paper falls into the mainstream of research examining the psychological determinants of professional involvement of nurses.

The main limitation of the present study is its cross-sectional character and the fact that that the studied positive orientation is a latent variable. The following question remains unanswered: To what degree does a positive orientation and professional burnout affect nurses’ life quality, which translates into the sphere of private and professional lives? Further research is necessary to verify the above question. For the purposes of further research, it is suggested that the selected sample takes into account the work-specific character across a wide variety of areas of the healthcare system.

### Implications for the Nursing Practice

The results of this study stress the essential need to protect nurses from the negative effects of professional burnout. Burnout manifests as physical and mental fatigue or exhaustion experienced by nurses. Protection requires the implementation of preventive programs that aim to deepen knowledge and help in the acquisition of competencies by nurses in recognizing and developing their own psychological resources, including positive orientation. Practical training should be included in prevention programs to shape and improve the ability to cope with stress. The current study recommends burnout prophylaxis programs as a permanent element of effective and efficient management in nursing and the interventions should include individual, group, and organizational activities in the work environment.

## 5. Conclusions

In light of research, a reduced level of positive orientation was proven to be the main determinant of professional burnout in Polish nurses. The most prominent predictive variable in identifying personal burnout was positive orientation, whereas in work-related burnout and burnout in contacts with patients, its predictive power was much weaker.

Financial situation, work experience, and coping strategies, such as cessation of actions, active coping, discharge, acceptance, and use of psychoactive substances were proven to be indicators of professional burnout with very low predictive power. They were found not to play a significant role in predicting personal burnout, work-related burnout, or burnout in contacts with patients.

A higher level of positive orientation was related to the use of adaptive coping strategies and was conducive to lowering the level of professional burnout for nurses in all three components.

With the increase in the level of positive orientation, a favorable change was observed in the aspect of the financial situation of nurses, whereas the age of the respondents and their work experience did not play a significant role in relation to positive orientation.

## Figures and Tables

**Figure 1 ijerph-16-04264-f001:**
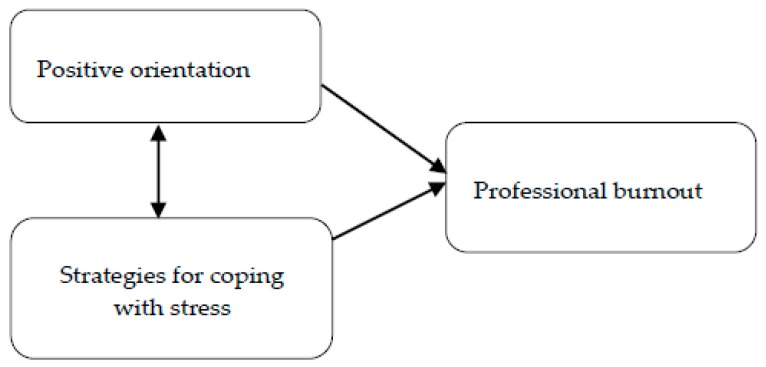
Model of the relationship between professional burnout, strategies for coping with stress, and a positive orientation.

**Table 1 ijerph-16-04264-t001:** Significance of impact: age, work experience, and financial situation on the level of positive orientation.

Positive Orientation Scale	Kruskal–Wallis Test (H)		Chi-Square Test (χ^2^)		Spearman’s Rank Correlation Coefficient (R)		
	H	*p*-Value	χ^2^	*p*-Value	R	t(N-2)	*p*-Value
Age	3.46	n.s.	7.87	n.s.	−0.008	−0.38	n.s.
Work experience	1.17	n.s.	11.26	n.s.	−0.02	−0.98	n.s.
Financial situation	152.83	0.001	155.23	0.001	0.28	12.45	0.001

*p* < 0.001; *p* < 0.01; *p* < 0.05.

**Table 2 ijerph-16-04264-t002:** Spearman’s rank correlation coefficient (R) between positive orientation and strategies for coping with stress.

Strategies for Coping with Stress (Mini-COPE)		Positive Orientation		
		Spearman’s (R)	T(N-2)	*p*-Value
1.	Active coping	0.22	9.71	0.001
2.	Planning	0.27	12.10	0.001
3.	Positive revalidation	0.25	11.09	0.001
4.	Acceptance	0.19	8.11	0.001
5.	Sense of humor	0.06	2.38	n.s.
6.	Turn to religion	0.03	1.12	n.s
7.	Seeking emotional support	0.27	12.01	0.001
8.	Seeking instrumental support	0.19	8.09	0.001
9.	Taking care of something else	0.06	2.50	0.01
10.	Denial	−0.10	−4.42	0.001
11.	Discharge	−0.09	−3.74	0.001
12.	Use of psychoactive substances	−0.09	−3.79	0.001
13.	Cessation of actions	−0.21	−8.96	0.001
14.	Self-blaming	−0.26	−11.63	0.001

*p* < 0.001; *p* < 0.01; *p* < 0.05.

**Table 3 ijerph-16-04264-t003:** Spearman’s (R) correlation coefficients between positive orientation, coping strategies, age, work experience, and burnout severity.

Variables		Personal Burnout		Work-Related Burnout		Burnout in Contacts with Patients	
		Spearman’s (R)	*p*-Value	Spearman’s (R)	*p*-Value	Spearman’s (R)	*p*-Value
**Positive orientation**		−0.39	0.001	−0.31	0.001	−0.30	0.001
Strategies for coping with stress (Mini-COPE)							
1.	Active coping	−0.17	0.001	−0.18	0.001	−0.15	0.001
2.	Planning	−0.13	0.001	−0.13	0.001	−0,13	0.001
3.	Positive revalidation	−0.08	0.001	−0.06	0.01	−0.07	0.001
4.	Acceptance	−0.09	0.001	−0.10	0.001	−0.10	0.001
5.	Sense of humor	0.03	n.s.	0.04	n.s.	0.03	n.s.
6.	Turn to religion	0.01	n.s.	−0.04	n.s.	−0.02	n.s.
7.	Seeking emotional support	−0.15	0.001	−0.13	0.001	−0.14	0.001
8.	Seeking instrumental support	−0.01	n.s	−0.04	n.s	−0.04	n.s
9.	Taking care of something else	0.05	0.05	0.03	n.s	0.03	n.s
10.	Denial	0.16	0.001	0.14	0.001	0.12	0.001
11.	Discharge	0.16	0.001	0.11	0.001	0.09	0.001
12.	Use of psychoactive substances	0.13	0.001	0.11	0.001	0.09	0.001
13.	Cessation of actions	0.20	0.001	0.18	0.001	0.17	0.001
14.	Self-blaming	0.19	0.001	0.11	0.001	0.10	0.001
Age		0.04	n.s.	0.10	0.001	0.06	0.01
Work experience		0.06	0.05	0.13	0.001	0.08	0.001
Financial situation		−0.24	0.001	−0.22	0.001	−0.21	0.001

*p* < 0.001; *p* < 0.01; *p* < 0.05.

**Table 4 ijerph-16-04264-t004:** Regression summary—personal burnout predictors.

Variables	*R* ^2^	β-Standardized	β	Error β	*t*	*p*-Value
Constant value			67.65	3.25	20.78	0.001
Positive orientation	0.16	−0.32	−1.22	0.09	−14.28	0.001
Financial situation	0.18	0.15	2.72	0.39	6.93	0.001
Discharge	0.19	0.10	2.41	0.55	4.38	0.001
Denial	0.20	0.07	1.53	0.49	3.14	0.001
Active coping	0.21	−0.06	−1.64	0.57	−2.88	0.001
Use of psychoactive substances	0.21	0.06	1.89	0.69	2.74	0.01
Work experience	0.21	0.05	0.09	0.04	2.60	0.01
R = 0.46; *R*^2^ = 0.21; Corrected *R*^2^ = 0.21						

Explanation: *p* < 0.001; *p* < 0.01; *p* < 0.05, R—correlation coefficient, *R*^2^—multiple determination coefficient, β—standardized regression coefficient, β—non-standardized regression coefficient, Error β—non-standardized regression coefficient error, *t—t*-test value.

**Table 5 ijerph-16-04264-t005:** Regression summary—predictors of work-related burnout.

Variables	*R* ^2^	β-Standardized	β	Error β	*t*	*p*-Value
Constant value			75.43	2.832	26.64	0.001
Positive orientation	0.10	−0.23	−0.85	0.086	−9.88	0.001
Financial situation	0.12	−0.15	−2.56	0.392	−6.53	0.001
Work experience	0.14	0.13	0.21	0.035	6.04	0.001
Cessation of actions	0.15	0.07	1.64	0.541	3.03	0.001
Active coping	0.15	−0.10	−2.47	0.570	−4.34	0.001
Discharge	0.16	0.08	1.93	0.527	3.67	0.001
R = 0.39; *R*^2^ = 0.16; Corrected *R*^2^ = 0.16						

*p* < 0.001; *p* < 0.01; *p* < 0.05.

**Table 6 ijerph-16-04264-t006:** Regression summary—predictors of burnout in contacts with patients.

Variables	*R* ^2^	β-Standardized	β	Error β	*t*	*p*-Value
Constant value			82.26	3.40	24.18	0.001
Positive orientation	0.09	−0.22	−0.97	0.10	−9.23	0.001
Financial situation	0.11	−0.14	−2.98	0.47	−6.33	0.001
Cessation of actions	0.12	0.07	2.00	0.65	3.07	0.001
Work experience	0.13	0.09	0.17	0.04	4.01	0.001
Active coping	0.14	−0.07	−2.23	0.71	−3.14	0.001
Discharge	0.14	0.07	2.03	0.64	3.16	0.001
Acceptance	0.14	−0.05	−1.39	0.65	−2.14	0.01
R = 0.37; *R*^2^ = 0.14; Corrected *R*^2^ = 0.14						

*p* < 0.001; *p* < 0.01; *p* < 0.05.

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
