# Peer review of "Positive Orientation and Strategies for Coping with Stress as Predictors of Professional Burnout among Polish Nurses"

_ijerph, 2019, doi:10.3390/ijerph16214264_

Round 1
Reviewer 1 Report
The reference 9 and 38 is the same. I think that in cite 9 they want to use this one "Gender, marital status and children as risk factors for burnout in nurses: a meta-analytic study".
Author Response
Thanks for your review.
Reference 38 was deleted.

Reviewer 2 Report
The authors considered suggestions ,and improved the manuscript.
I have only a suggestion:
Authors should delete the infomation " Statistically significant" under each table is redundant.
Author Response
Thanks for your review.
Deleted.

This manuscript is a resubmission of an earlier submission. The following is a list of the peer review reports and author responses from that submission.
Round 1
Reviewer 1 Report
The manuscript addresses an important topic, investigating predictors of professional burnout among nurses . The manuscript is well written. I have some considerations :
The introduction section is quite long, should be shortened in my opinion.
The references , if possible , should be updated. Almost no work of the last two years is present.
Methods: the study design must to be made explicit. This is a cross sectional study.Cross sectional study should be structured according to Strobe Statement . Please verify if the study meets the strobe criteria and cite as: "Vandenbroucke JP, von EE, Altman DG, Gotzsche PC, Mulrow CD, Pocock SJ, et al. Strengthening the reporting of observational studies in epidemiology (STROBE): explanation and elaboration. PLoS Med. 2007;4(10):e297". the study involves midwives and nurses but how many nurses and how many midwives? In statistical analysis why the authors do not take into account differences between these two groups?
Authors never mentioned the wards were the respondents came from. This, especially for burnout , is an essential topic. It is known that in certain department (es psichiatry or emergency) the risk of burnout is definitely higher .
Some other detail about the administration of the questionnaire should be mentioned. Once administered, the questionnaire were left a few day or were completed at the same time? Who distribuited the questionnaire?The return percentage varied 25.3% to 87.5. Do you have explanation for these different results? Furthermore, response rate of 62.6% is low please insert this point in limitation of study. The authors be certain that the responding participants did not have a bias?
A section with limits and strenghts of the work should be added.
In conclusion section future direction and implications of the study should be added.
Reviewer 2 Report
The study determines the role that positive guidance and coping strategies have on burnout in nurses. It is a very interesting article and a wide-reaching topic. However, prior to its publication, certain parts need clarification.
Correct the impersonal style, “We….”
In section 2.1, it is necessary to explain the procedure for selecting the participants for the study. How were they selected? Who administered the questionnaires? One important piece of information when describing the participants would be their stability over time, would it be possible to add this information to the study?
In the results section, lines 246-247, it is not necessary to add any data that is not significant.
The clarification that appears below the tables “Statistically significant….” should be clarified in the table, otherwise it does not make sense.
Eliminate p< 0.0001 and substitute p≤0.001 for it.
Include the level of significance in the analysis in table 5.
I would recommend updating the bibliographic references, as they should include more recent studies from within the last 5 years.
Reviewer 3 Report
Dear authors, the study topic is very interesting and it analyze a big problem for nurses, burnout. I have some recommendations for its improvement and some doubts
Abstract
The abstract is clear and easy to understand and it show the main results. I would change just one thing. In line 20 the authors says "Most nurses reported..." but I think that the 27% is not most of the nurses. I would say "the 27% of the nurses reported work-related burnout...".
Introduction
The introduction include a good analysis of the topic based on the current literature.
Along the text the authors use a wrong style of citation like line 97 "C.S Carver et al." should be "Carver et al." or like line 101 "Ch. Maslach" it should be just "Maslach". Please revise the text, the name capitals letters are not included in text citations.
In line 122 I would delete the phrase "for the authors of this study"
Methods
Before the section 2.1 I would include the type of study
I feel that is not clear the use of this tests "The Kruskal-Wallis (H) test, to compare the distribution of the variables for the number of trials >2, the post-hoc test for multiple mean ranks for all trials". I do not understand what do you mean when you say "trials"
Results
Why 3 different statistical tests are used for the same variables analysis? Table 1. Which "independent groups" were used for those analysis? it is not easy to understand.
The results should be divided in more paragraphs depending on the topic. It is very hard to follow the reading of the text.
The text that refers to one table should all be placed before the table.
Discussion and Conclusion
The discussion should be divided in more paragraphs depending on the topic
I would include information about the influence of personality factors in nurses burnout. You talk about some psychological factors in the discussion but there is nothing about the nurses personality factors and burnout. Personality can also influence the positive orientation.
A limitations of the study paragraph must be included.
The conclusion should not have numbers to differentiate each paragraph.